# The Effect of Sucrose Supplementation on the Micropropagation of *Salix viminalis* L. Shoots in Semisolid Medium and Temporary Immersion Bioreactors

Diego Gago [1,2], Saladina Vilavert [1], María Ángeles Bernal [2], Conchi Sánchez [1], Anxela Aldrey [1] and Nieves Vidal [1,*]

1   Instituto de Investigaciones Agrobiológicas de Galicia, Consejo Superior de Investigaciones Científicas, Apdo 122, 15780 Santiago de Compostela, Spain; tresdedecembro@gmail.com (D.G.); salayvilavert28@gmail.com (S.V.); conchi@iiag.csic.es (C.S.); anxela.aldrey@iiag.csic.es (A.A.)
2   Departamento de Biología, Facultad de Ciencias, Universidade da Coruña, Campus da Zapateira s/n, 15071 A Coruña, Spain; angeles.bernal@udc.es
*   Correspondence: nieves@iiag.csic.es

**Abstract:** The effect of sucrose concentration on the micropropagation of axillary shoots of willow was investigated. The following factors were examined: the culture system (semisolid medium in glass jars versus liquid medium in temporary immersion bioreactors), the type of explant (apical and basal sections), the frequency of immersion, and $CO_2$ enrichment. Shoots and leaf growth were significantly higher in RITA® bioreactors than in the jars for all the sucrose treatments. Apical or basal sections of willow cultured in bioreactors under high light intensity (150 µmol m$^{-2}$ s$^{-1}$) and ventilated six times a day with $CO_2$-enriched air were successfully proliferated without sucrose, whereas shoots cultured in jars did not proliferate well if sucrose concentration was 0.5% or lower. More roots were formed when sucrose was added to the medium. Shoots cultured in bioreactors were successfully acclimatized irrespective of the sucrose treatment and the root biomass when transferred to ex vitro conditions. This is the first report of photoautotrophic willow micropropagation, our results confirm the importance of proper gaseous exchange to attain autotrophy during in vitro propagation.

**Keywords:** acclimatization; autotrophy; liquid medium; photosynthesis; RITA®; ventilation; willow

## 1. Introduction

There are both economic and physiological advantages to the use of bioreactors for micropropagation of many plants including easily propagated plants such as willows [1–6].

The cost of gelling agents is eliminated, automation can be implemented, and when bioreactors incorporate aeration, the increased uptake of nutrients via the liquid medium together with the renewal of the air inside the bioreactors can improve the physiological state of the explants as well as shortening the plant propagation cycle. While the initial investment can be high [5], the increased survival and vigour of in vitro plantlets when transferred to ex vitro conditions can offset the costs, depending on the commercial or ecological value of the obtained plants.

In vitro plant cultures are not normally autotrophic as they have a need for carbohydrates as a carbon and energy source, which is most commonly sucrose. Sucrose can act as a signaling molecule and maintains osmotic potential in cells, but high concentrations decrease net photosynthesis and can cause abnormalities [7]. It has been reported that the photosynthetic ability of tissues cultured in bioreactors with forced ventilation is enhanced relative to other methods of in vitro culture [2], creating the possibility for decreasing or even eliminating the conventional sugar supplementation, producing photoautotrophic cultures [7].

In this study, we explored the effect of sucrose supplementation during the micropropagation of a willow species (*Salix viminalis* L.), with shoots cultured in semisolid medium and in temporary immersion bioreactors.

Willows (genus *Salix*) comprise approximately 400 species of deciduous trees and shrubs belonging to the Salicaceae family, the same as poplars, aspen and cottonwoods. They are widely distributed from subtropical to cold climates of the Northern Hemisphere and have been cultivated for centuries around the world. Willows are multipurpose trees, with traditional uses ranging from timber, fencing, foraging and basket work to charcoal manufacture. They are also used for boxing, pulp and fibre board production, for controlling erosion on river and stream banks, for phytoremediation and as an energy crop mostly for municipal use producing heat and power [8,9]. *S. viminalis*, also known as basket willow or osier willow, is (like most willows) a dioecious species, insect and wind pollinated. In nature, it propagates sexually by seeds that are dispersed by wind or water, and asexually through the rooting of broken twigs and branches. This rooting ability enabled its extensive historical cultivation since ancient times. Its fast initial growth rate, perennial habit and ability to repeatedly regrow vigorously from coppiced stools, made it one of the most suitable species for the short rotation coppice for bioenergy production [9,10]. In addition, it is a known hyperaccumulator of metals and organic substances, making it a prime candidate for phytoremediation [11]. For these reasons, S. viminalis has been widely used in breeding programmes [9,10,12]. Good breeding efficiency requires seedlings from controlled crossings to be rapidly multiplied to produce sufficient clonal plantlets for replicated trials in a range of environments. *S. viminalis* roots easily, but it is often an advantage to accelerate plant multiplication of selected genotypes required for testing or distribution [12]. Some micropropagation protocols in a semisolid medium have been developed for *S. viminalis* and other willow species [12] and references therein. In our laboratory we explored its use as a model plant for our research in a liquid medium culture. We used the temporary immersion system Plantform™ and RITA® bioreactors and obtained higher proliferation rates than in a semisolid medium [6].

The aim of the present study was to use *S. viminalis* as a model to develop an efficient protocol for culturing axillary shoots of woody plants without sucrose supplementation. Variables such as growth conditions (light intensity and $CO_2$ enrichment), frequency of immersion, explant type and sucrose supplementation were evaluated in relation to shoot quality, photosynthetic pigment content, in vitro proliferation and ex vitro performance.

## 2. Materials and Methods

### 2.1. Plant Material and Micropropagation in Semisolid and Liquid Medium

*Salix viminalis* shoot cultures were previously established in vitro from a mature tree [6]. Shoots were maintained in 75 mL of semisolid medium (SSM) in 500 mL glass jars (6 explants per jar, i.e., 12.5 mL of medium per explant) and were subcultured every 8 weeks. The jars were sealed with plastic film (Figure 1a) and did not receive forced ventilation. The SSM consisted of an MS (Murashige and Skoog) [13] salt and vitamin mixture with half-strength nitrates (MS-$\frac{1}{2}$N) and supplemented with 0.22 μM of BA, 3% sucrose and 0.65% Plant Propagation Agar (Condalab, Madrid, Spain). The medium was adjusted to pH 5.7 before being autoclaved at 120 °C for 20 min. Cultures were incubated under a 16-h photoperiod provided by cool-white fluorescent lamps (photosynthetic photon flux density (PPF) from 50–60 μmol m$^{-2}$ s$^{-1}$) at 25 °C light/20 °C dark.

Shoots were also cultured by temporary immersion (TIS) in a liquid medium in commercial RITA® (Vitropic, Saint Mathieu de Tréviers, France) bioreactors (Figure 1b) as described by Regueira et al. [6]. Liquid medium (LM), of the same composition as SSM but devoid of agar, was autoclaved before being added to the containers. In RITA® bioreactors, the medium reaches the explants when forced ventilation with filtered air is applied at fixed intervals [1]. Each RITA® contained 12 explants and 150 mL of LM (12.5 mL of medium per explant). Explants were subcultured every 8 weeks as described above for the explants cultured in jars.

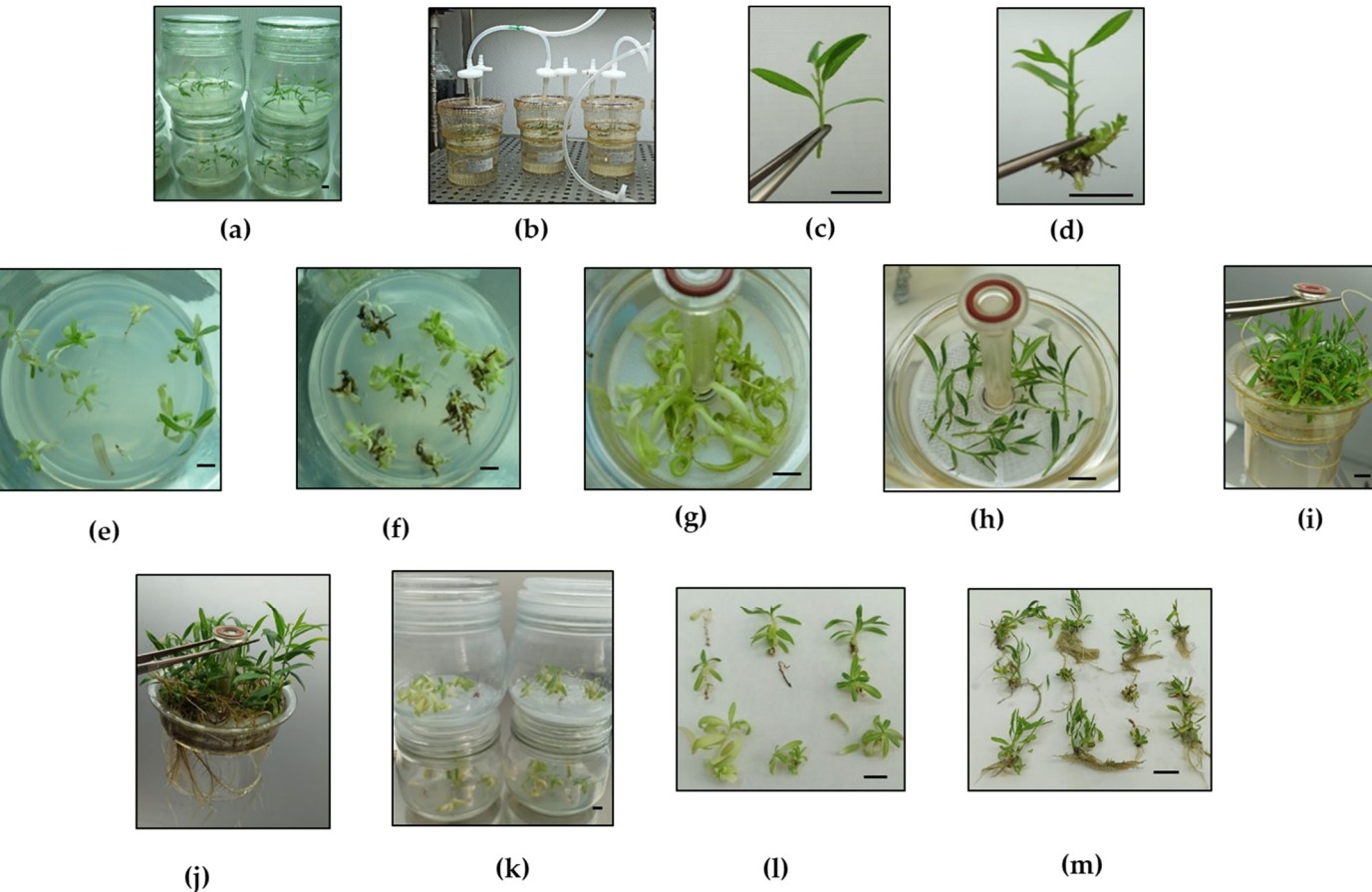

**Figure 1.** Micropropagation of willow in semisolid medium and bioreactors with different sucrose supplementation. (**a**) Jars used for propagation in semisolid medium. (**b**) RITA® bioreactors used for propagation in liquid medium by temporary immersion. (**c**,**d**) Apical (**c**) and basal (**d**) sections of willow used as initial explants. (**e**,**f**) Apical (**e**) and basal (**f**) sections of willow cultured in glass jars without sucrose in standard conditions. (**g**) Apical sections of willow cultured in RITA® vessels without sucrose and with 3 immersions/day in standard conditions. (**h**–**j**) Apical segments at time zero (**h**) and after 8 weeks of cultivation under PAM conditions and 6 immersions/day in RITA® vessels without sucrose (**i**) or with 3% sucrose (**j**). (**k**–**m**) Apical sections cultured in jars under high light intensity without sucrose (**k**,**l**) or with 3% sucrose (**m**). Bars 10 mm.

The following experiments were carried out to test the effects of variables such as growth conditions, frequency of immersion, explant type, sucrose supplementation and culture system. Two or four RITA® vessels and four or eight jars per treatment were used in micropropagation experiments, and sixteen plantlets per treatment in acclimatization experiments.

*2.2. Growth Conditions and Frequency of Immersion*

Two experiments were carried out while studying these variables. In the first one, willow apical and basal sections (20–25 mm) with 4–5 leaves (Figure 1c,d) were excised from shoots growing in jars with 3% sucrose. These initial explants (48 explants per treatment) were cultured in jars or RITA® vessels without sucrose under standard (ST) conditions (conventional PPF (~50 $\mu$mol m$^{-2}$ s$^{-1}$) and ambient $CO_2$ (~400 ppm)). Explants cultured in RITA® were immersed 3 times per day. We used 20–25 mm explants, longer than the 15 mm sections we had used in our previous study under photomixotrophic conditions [6], as increasing the explant size and leaf area was beneficial for other plants like potato when cultured without sucrose [14].

In the second experiment, apical sections (48 explants per treatment) were cultured in RITA® and were immersed in LM supplemented with 1 or 3% sucrose for 1 min 6 or 16 times per day (every 1.5 or 4 h). The explants were cultured under ST conditions or

in an experimental unit previously designed for chestnut photoautotrophic micropropagation [15]. In this experimental unit, designated as PAM, the cultures grew under high PPF (150 $\mu$mol m$^{-2}$ s$^{-1}$), and $CO_2$-enriched air (~2000 ppm) was injected to the bioreactors during immersion. The photoperiod and temperature regime were the same as under ST conditions.

### 2.3. Culture System and Sucrose Supplementation

Apical sections of willow were cultured in RITA® and glass jars under 150 $\mu$mol m$^{-2}$ s$^{-1}$ PPF in the PAM experimental unit, with LM or SSM supplemented with 0, 0.5, 1 or 3% sucrose. Four jars and two RITA® vessels were used for each treatment and replication. Explants cultured in temporary immersion systems were immersed for 1 min every 4 h, and received $CO_2$-enriched air as described above, whereas the explants cultured in jars did not receive forced aeration. Two successive subcultures were carried out to minimise any possible carryover of sugars (and other growth factors) present in the tissues of the initial explants. In the first subculture, all the explants were obtained from shoots cultured in SSM with 3% sucrose and placed in jars or bioreactors with different sucrose concentrations (8 treatments), whereas in the second subculture the initial explants for each treatment were obtained from shoots previously developed in that specific treatment. The whole procedure was repeated twice, and only shoots from the second series of each repetition were used to elaborate the results.

### 2.4. Explant Type

Two types of explant were evaluated: 20–25 mm apical or basal sections of willow shoots, with 4–5 leaves (Figure 1c,d). These explants were cultured in RITA® (two vessels per treatment) under 150 $\mu$mol m$^{-2}$·s$^{-1}$ PPF. For immersion in the liquid medium, $CO_2$-enriched air was applied for 1 min every 4 h.

### 2.5. Acclimatization

Shoots grown in RITA® with 0, 0.5, 1 or 3% sucrose were extracted from the bioreactors and the roots formed spontaneously were uniformly cut to 30 mm lengths from the base of the shoots. The plantlets (16 per treatment) were transferred to plastic plug trays (size of the plugs 52 × 52 mm, 60 mm height) filled with a peat:perlite (3:1) mixture and placed in a controlled environmental chamber (Fitotron SGC066, Sanyo Gallencamp PLC, Leicestershire, UK) with a photoperiod of 16 h light: 8 h dark, a photosynthetic photon flux from 240–250 $\mu$mol m$^{-2}$·s$^{-1}$, a temperature of 25 °C (day) and 20 °C (night), and relative humidity of 85%.

After 3 weeks in the phytotron, plantlets were transferred to 1.3 L pots and placed into a greenhouse. The percentage of survival and plantlet growth were recorded when transferred to the greenhouse and 2 weeks later.

### 2.6. Data Recording and Statistical Analysis

The following dependent variables were determined to evaluate the aforementioned parameters during the micropropagation and acclimatization stages:

Micropropagation: (1) number of shoots longer than 20 mm produced by each explant (NS); (2) the multiplication coefficient (MC), defined as the number of new 20 mm nodal segments produced from an initial explant; (3) length of the longest shoot per explant (SL); (4) length and width of the largest leaf per explant (LL and LW), in order to calculate leaf area (LA) as $\frac{1}{2}$ LL*LW; (5) number of rooted shoots; (6) fresh and dry mass of shoots and roots; (7) pigment concentration of the two uppermost fully expanded leaves per explant (chlorophyll a, b and carotenoids were extracted with dimethylformamide and quantified following the method described by Wellburn [16]).

Acclimatization: (8) Survival percentage; (9) Shoot length; (10) leaf area of the largest leaf, calculated as detailed in 4. These parameters were recorded at the moment of transfer

to ex vitro conditions in a phytotron (T1), 3 weeks later when transferred to a greenhouse (T2), and 2 weeks afterwards (T3).

Data were collected from 24–48 explants per treatment (two or four RITA® vessels and four or eight jars per treatment) in micropropagation experiments, and to 16 plantlets per treatment in acclimatization experiments. The data were analysed by Levene's test (to verify the homogeneity of variance) and the Shapiro–Wilk test of normality. The data were then subjected to analysis of variance (ANOVA), followed by the comparison of group means (Tukey-b test), or to the Welch ANOVA, followed by Games–Howell post-hoc comparison (when heteroscedasticity was detected). When an interaction between two factors was indicated by the two-way ANOVA, Bonferroni's adjustment was applied to detect the simple main effects in multiple post-hoc comparisons. Statistical analyses were performed using SPSS 26.0 (IBM) [17].

## 3. Results

### 3.1. Growth Conditions and Frequency of Immersion

Our first attempts to culture willow explants without sucrose in ST with 3 immersions per day were unsuccessful, both in jars and in bioreactors. As shown in Figure 1e–g, only small shoots with chlorotic or hyperhydric leaves were obtained in both systems. To find more appropriate conditions to culture willow with low sucrose, we performed another experiment with bioreactors comparing two environmental conditions (ST and PAM), two frequencies of immersion (6 and 16 per day), and two sucrose concentrations (1 and 3%).

The results of this experiment can be observed in Figure 2. Data were analyzed separately for each sucrose supplementation. The number of shoots was significantly affected by the growth conditions ($p < 0.001$ for both sucrose concentrations), with the highest proliferation obtained in PAM in both cases (Figure 2a). Although increasing the frequency of immersion from once every 4 to once every 1.5 h a day did not affect the total number of shoots (Figure 2a), many of them (more than 65%) were hyperhydric and not suitable for propagation. An immersion interval of 1.5 h led to a significant decrease in SL (Figure 2b, $p < 0.001$ in both sucrose treatments).

Significant interaction between growth conditions and the frequency of immersion were detected for SL and for the photosynthetic pigments (Figure 2b–d; $p$ values ranging from less than 0.001 to 0.040), except for carotenoids in shoots treated with 1% sucrose ($p = 0.071$). In general, for shoots cultured with low sucrose concentration the best results were obtained in PAM and with one immersion every 4 h, so these conditions were applied in the next experiments.

### 3.2. Culture System and Sucrose Supplementation

Figure 3 shows the results of culturing apical sections of willow in RITA® and glass jars under high PPF and with $CO_2$ supply (in the case of bioreactors). Proliferation parameters (SL, MC and LA) were significantly higher in TIS than in SSM irrespective of the sucrose supplementation (Figure 3a–c). Regarding the effect of sucrose, vigorous growth was obtained in RITA® for all sucrose treatments (Figures 1h–j and 3a,b), whereas shoots cultured in jars with low sucrose showed reduced SL and MC (Figures 1k,l and 3a,b).

There was a statistically significant interaction between the culture system and sucrose supplementation on all the parameters measured ($p \leq 0.001$), meaning that sucrose concentration affected growth and pigments in a different way if the explants were cultured in SSM or TIS. In jars, shoots grew better with 3% sucrose (Figure 1m), whereas in RITA®, longer shoots with more nodes and larger leaves were obtained without exogenous sugar (Figure 3a–c). In jars, many of the leaves with reduced sugar supplementation were chlorotic (Figure 1k,l), and this was reflected in their reduced pigment content compared to shoots cultured in RITA® or in SSM with 3% sucrose (Figure 3e,f). Similar pigment concentrations were found in the four RITA® treatments and in glass jars with 3% sucrose (Figure 3e,f), whereas SL and MC showed some significant differences between these five treatments, being most evident in the case of jars (Figure 3a,b).

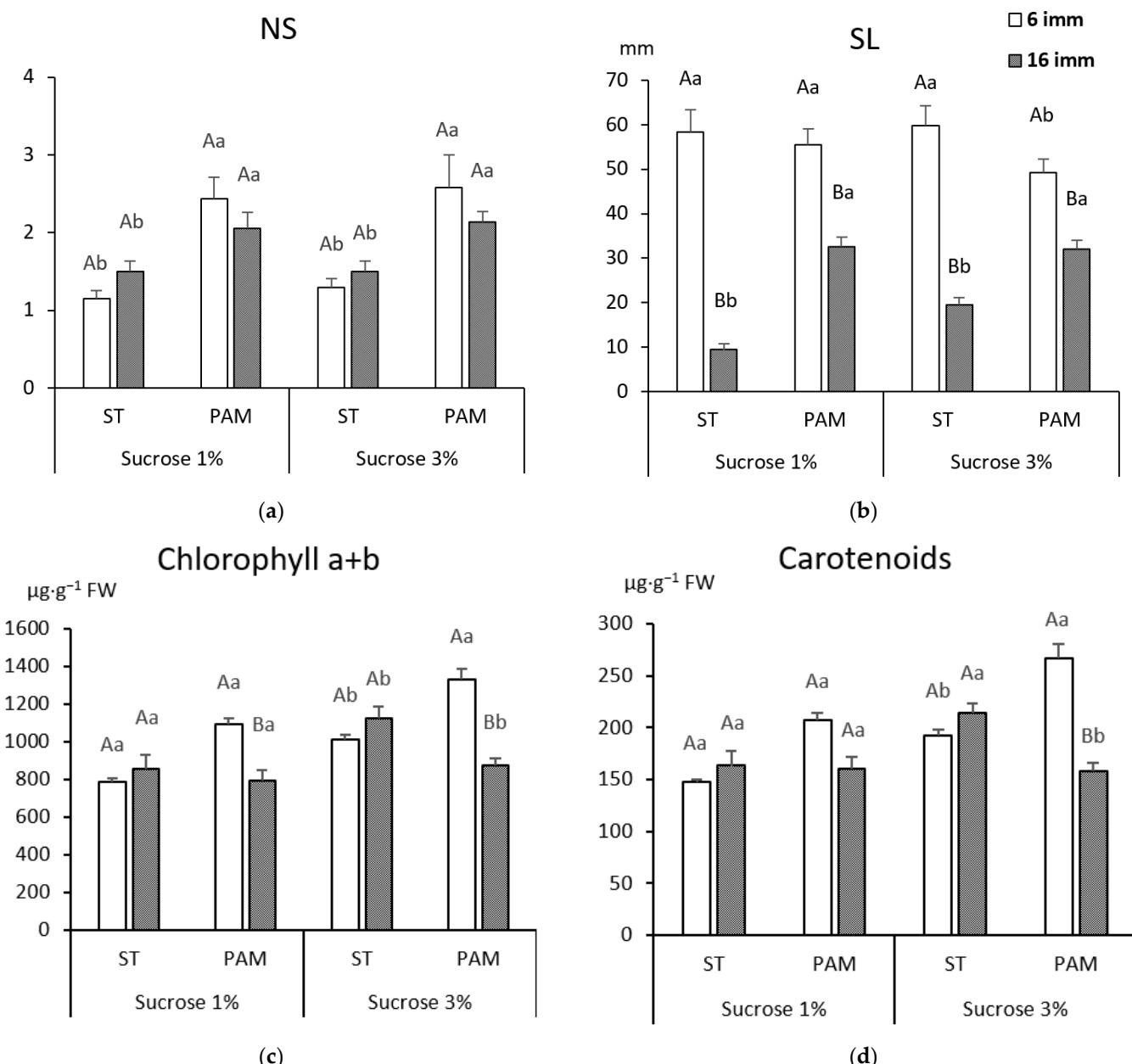

**Figure 2.** Effect of sucrose supplementation, immersion frequency (6 or 16 immersion/day) and growth conditions (standard or PAM) on proliferation rates (**a**,**b**) and pigment content (**c**,**d**) of apical sections of willow shoots cultured in RITA® vessels. Values are the mean ± standard error from two replicate trials each with 2 RITA® vessels. For each sucrose supplementation and variable, different capital letters indicate significant differences ($p < 0.05$) in relation to the frequency of immersion, and different lowercase letters indicate significant differences in relation to growth conditions (standard or PAM). Due to the significant interaction found in SL and pigments, Bonferroni's adjustment was applied to detect simple main effects. FW Fresh weight, NS number of shoots per explant, SL length of the longest shoot, ST standard conditions, PAM high PPF + $CO_2$-enriched air.

Willow shoots formed adventitious roots naturally during the multiplication step; the extent of root formation was dependent on the sucrose supplementation for both jars and bioreactors. The percentage of rooted shoots decreased sharply in jars cultured with less than 1% sucrose (Figure 3d). In bioreactors almost all the shoots cultured in RITA® developed roots, but their number, length and branching decreased when low sucrose concentrations were used. Since roots grew through the gaps in the bioreactor baskets, preventing the collection of data for individual shoots, we recorded the shoot and root biomass of each RITA® as a whole (Figure 4). Willow shoots cultured photoautotrophically

in a medium devoid of sucrose exhibited the highest amount of aerial biomass, but the lowest root biomass, which increased when sucrose was added to the medium.

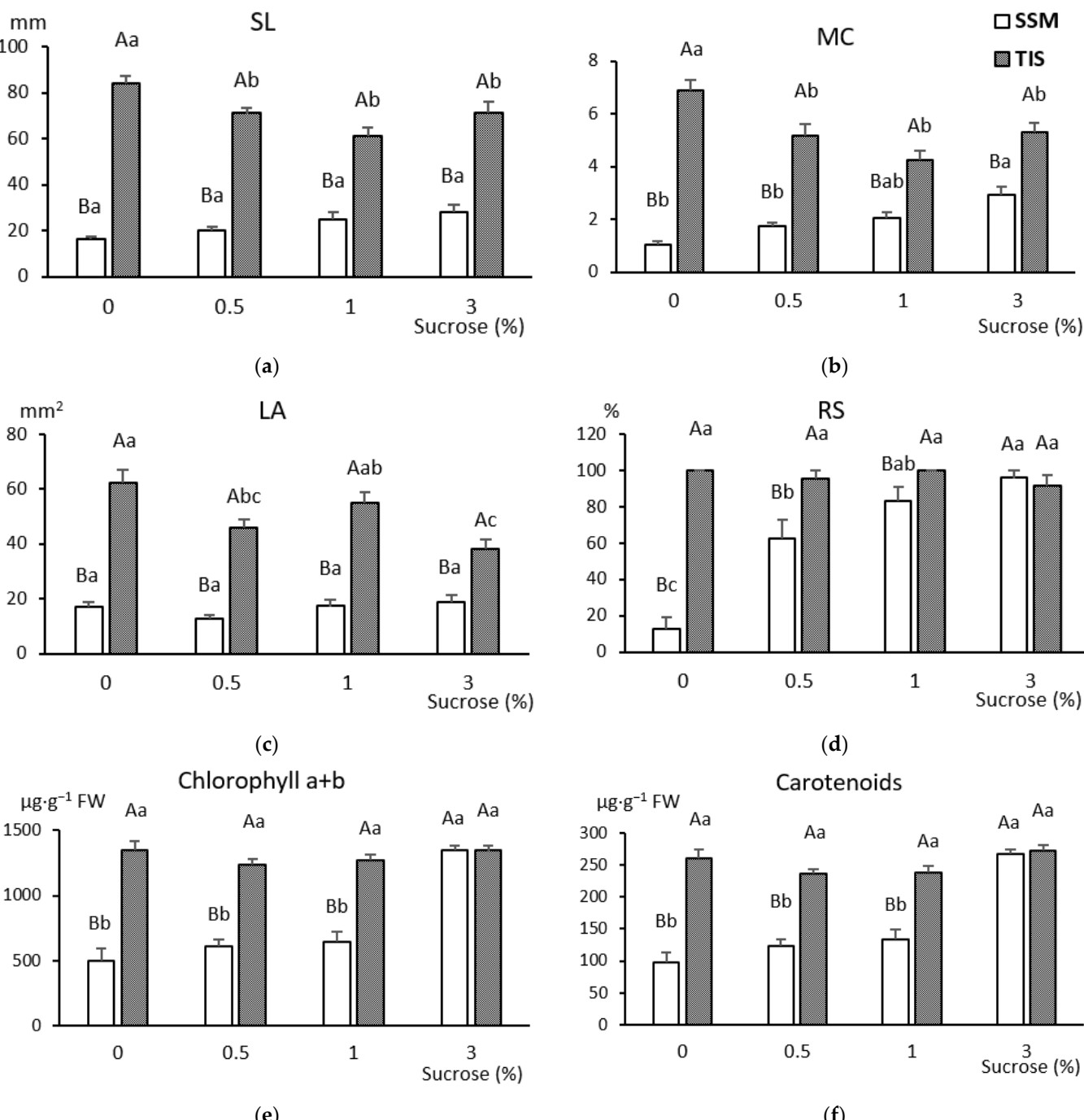

**Figure 3.** Effect of culture system and sucrose supplementation on proliferation rates (**a**–**d**) and pigment content (**e**,**f**) of apical sections of willow shoots. All explants were grown under 150 μmol m$^{-2}$s$^{-1}$ PPF, and in the case of bioreactors $CO_2$-enriched air (2000 ppm) was supplied during immersion (1 min every 4 h). Values are the mean ± standard error from two replicate trials each with 4 jars and 2 RITA$^{®}$ vessels. Percentage data were subjected to arcsine transformation prior to analysis and untransformed data are presented in the graphics. For each variable, different capital letters indicate significant differences in relation to culture system (SSM or TIS), and different lowercase letters indicate significant differences in relation to the sucrose supplementation ($p < 0.05$). When significant interaction between factors occurred, a Bonferroni adjustment was made to detect simple main effects between means. FW Fresh weight, SL length of the longest shoot, MC multiplication coefficient, LA leaf area of the largest leaf per explant, RS percentage of rooted shoots.

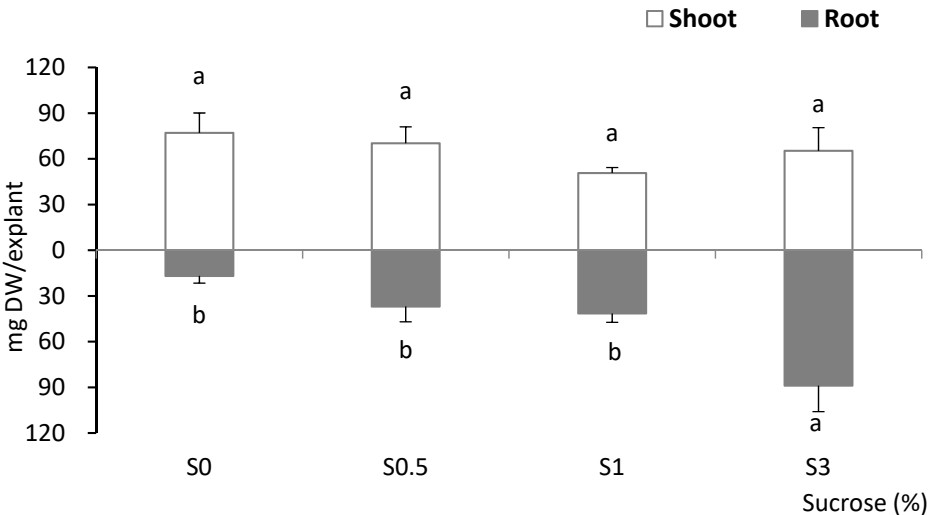

**Figure 4.** Effect of sucrose supplementation on shoot and root dry weight of apical sections of willow grown in RITA®. Explants were cultured under 150 $\mu$mol m$^{-2}$·s$^{-1}$ PPF and $CO_2$-enriched air (2000 ppm) was supplied during immersion (1 min every 4 h). Values are the mean $\pm$ standard error from two replicate trials each with 2 RITA® vessels. For each variable (shoot or root weight) different letters indicate significant differences in relation to the sucrose supplementation ($p < 0.05$). DW Dry weight.

### 3.3. Explant Type

The apical and basal sections of willow were cultured in bioreactors with a medium devoid of sugar (Figure 5). Basal sections showed significantly more proliferation capacity than apical sections, and this was more evident for the treatment without sucrose (Figure 5).

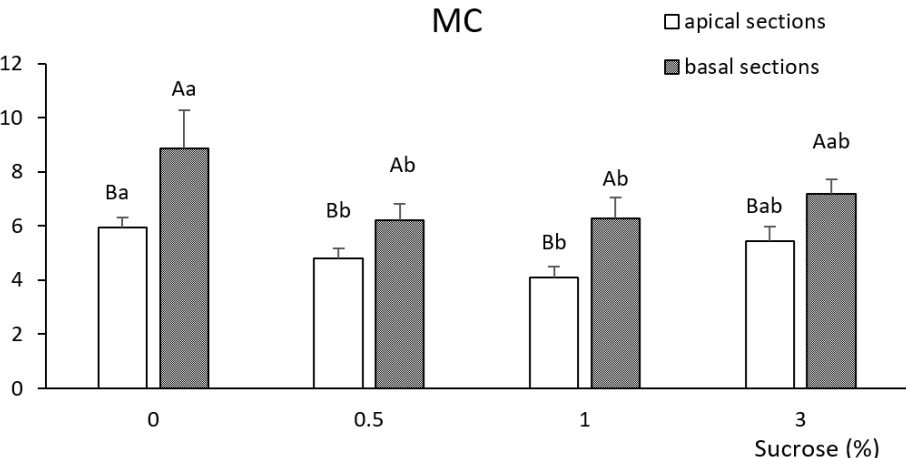

**Figure 5.** Effect of explant type and sucrose supplementation on multiplication coefficient of willow. Apical and basal sections were grown in RITA® under 150 $\mu$mol m$^{-2}$·s$^{-1}$ PPF. For immersion in liquid medium, $CO_2$-enriched air was applied for 1 min every 4 h. Values are the mean $\pm$ standard error from 2 RITA® vessels. Different capital letters indicate significant differences in relation to explant type, and different lowercase letters indicate significant differences in relation to the sucrose supplementation ($p < 0.05$). MC—multiplication coefficient.

### 3.4. Acclimatization

Spontaneously rooted shoots cultured in bioreactors with 0, 0.5, 1 or 3% sucrose were transferred to substrate for acclimatization (Figures 6 and 7).

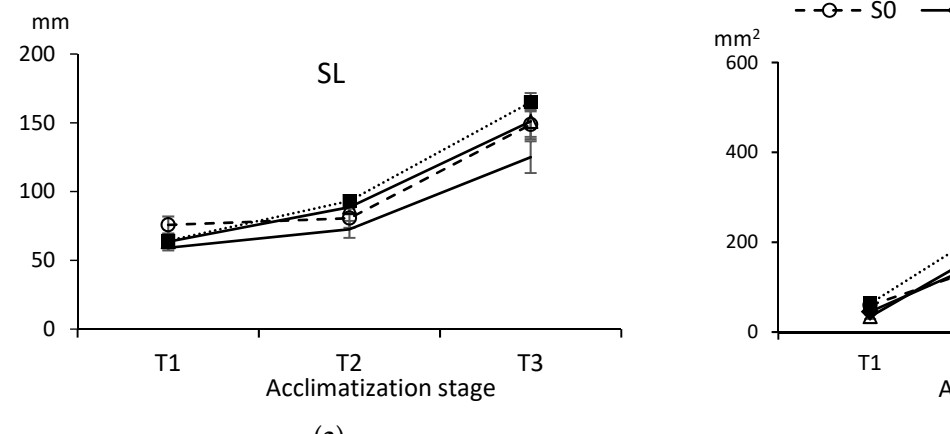

(**a**)                                                                                                   (**b**)

**Figure 6.** (**a**) Shoot length and (**b**) leaf area of willow plantlets during 3 stages of acclimatization (T1–T3). Rooted shoots cultured in RITA® with different sucrose concentrations under PAM conditions were measured when transferred to the phytotron (T1), to the greenhouse 3 weeks later (T2), and 2 weeks afterwards (T3). Values are the mean ± standard error from two replicate trials each with 8 shoots per treatment. SL shoot length; LA leaf area of the largest leaf per explant; S0, S0.5, S1 and S3 indicate sucrose supplementation of 0, 0.5, 1 and 3%, respectively.

Shoots grown in jars were not included as the roots developed in SSM with 0.5% sucrose or less were scarce (Figure 3d) and barely longer than 8–10 mm, whereas the shoots were too weak and chlorotic to undergo acclimatization (Figures 1k,l and 3a,e,f). For shoots cultured in TIS, survival was high for all treatments, irrespective of having shown more or fewer roots (Figure 4) or longer or smaller shoots and leaves (Figure 3a,c) during the multiplication stage. Only 3 plantlets of 64 (2 of S0.5 and 1 of S3) died during the first days after transplanting. The size of the shoots and leaves at the moment of the transference to phytotron was significantly different among treatments (*p* = 0.042 and 0.002, respectively), as could be inferred for the results shown in Figure 3. However, 3 and 5 weeks after being extracted from the bioreactors no significant differences were observed for any of the parameters, with *p* values of 0.092 and 0.079 for SL (Figure 6a) and 0.096 and 0.575 for LA (Figure 6b).

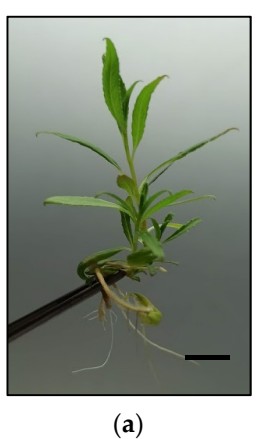   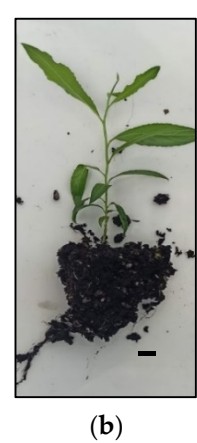   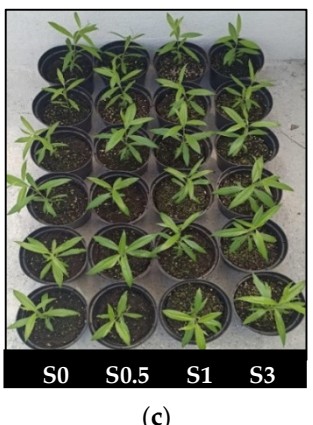   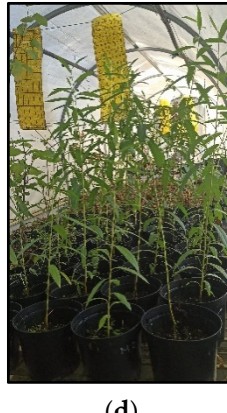

(**a**)                          (**b**)                          (**c**)                          (**d**)

**Figure 7.** Acclimatization of willow shoots micropropagated in bioreactors in PAM conditions. (**a**,**b**) Apical sections cultured without sucrose at the moment of transfer to ex vitro conditions (**a**) and after 3 weeks in a phytotron (**b**). Plantlets cultured with 0, 0.5, 1 and 3% sucrose after 3 weeks in a phytotron and 2 weeks in a greenhouse (**c**). Plantlets after 9 weeks in the greenhouse (**d**). Bars 10 mm.

The aspects of the plantlets during acclimatization are shown in Figure 7. Shoots cultured without sucrose grew successfully after transplanting and rapidly increased their root system (Figure 7a,b), despite showing a lower root biomass during the in vitro propagation stage. Five weeks after transplanting all the plantlets showed a similar

appearance (Figure 7c), and 7 weeks later they had reached 60–70 cm, irrespective of the treatment (Figure 7d).

## 4. Discussion

Since Murashige and Skoog [13] described their MS medium in 1962, 3% sucrose has been the most used carbohydrate for photomixotrophic micropropagation. However, in recent decades some researchers have claimed decreasing sucrose concentration in the medium or eliminating it completely. This could benefit in vitro propagation by enhancing photosynthesis and promoting a more natural physiology of the propagated plants [7]. The aim of this study was to investigate the possibility of culturing willow shoots without sucrose, both in SSM and in LM in bioreactors.

In our previous study on this species [6], we had explored the feasibility of lowering the sucrose content during part of the subculture period. We used shoots cultured in jars and RITAs® (3 immersions/day) in ST (low light intensity, and normal air supply for the bioreactors), and we did not observe any detrimental effect after reducing the sucrose content to 1% during the last 2 weeks of the 6-week subculture period [6]. However, in the present study we could not propagate shoots cultured under ST conditions with SSM or LM completely devoid of sucrose for the whole culture period, indicating that in those conditions the physiological improvement claimed by using TIS [18,19] was insufficient to sustain willow photoautotrophic growth.

We carried out several experiments aimed at finding the appropriate conditions to propagate willow plants without sugar. In these experiments we tested the effect of modifying the frequency of immersion and the growth conditions (light intensity and $CO_2$-enrichment) of shoots cultured in bioreactors and compared their performance with shoots cultured in closed systems (jars).

A high immersion frequency can enhance growth as it increases nutrient absorption and gas exchange [1], but the optimal value of this parameter is quite variable among plants. In other woody species, such as apple, eucalyptus and hazelnut, increased immersion frequency improved yield and plant quality: in RITA®, shoot number and length increased when apple rootstock M26 shoots were immersed every 1.5 h [20] and the highest multiplication in eucalyptus was achieved using 30-s flushes of medium every 10 min [18], whereas in a Liquid Lab Rocker™ bioreactor [21], immersion intervals of 25 s were suitable to culture hazelnut shoots. However, in our study, an immersion interval of 1.5 h (16 times per day) led to a significant increase of hyperhydricity and a decrease in shoot length. Similar detrimental effects of increasing the immersion frequencies were observed in carnation and *Colocasia esculenta* L. when this parameter was set to 1 or 2 h intervals instead of 4–8 h [22,23]. In willow, 6 immersions per day produced good quality shoots and were used in the rest of the experiments. Immersion every 3–6 h has been successfully used for other tree species as *Crescentia cujete* L. [24], *Fraxinus mandshurica* Rupr. [25], *Tectona grandis* L. [26], *Ilex paraguariensis* A.St.-Hil. [27] and apple [28].

Regarding the growth conditions, PPF of approximately 150 µmol m$^{-2}$s$^{-1}$, and forced ventilation with 2000 ppm $CO_2$-enriched air proved favourable for culturing willows in TIS with 1% sucrose (Figure 2), and these conditions were used in the experiments aimed at eliminating sucrose from the culture medium. When we compared shoots cultured in RITA® under these conditions with shoots cultured in closed jars under the same PPF the values obtained in TIS were always superior to those observed in SSM (Figure 3). Better performance of explants cultured by TIS was reported for trees like eucalyptus, red cedar, teak, pistachio, chestnut, hazelnut, yerbamate and olive (reviewed in [5]) but, in most of those studies, sugar was added to the culture medium. Willow shoots cultured in bioreactors produced high proliferation for all the sucrose treatments. In TIS, the best results occurred with shoots cultured without sucrose, whereas in jars the opposite trend was noted and no sustainable growth occurred with sucrose concentrations of 0.5% or less.

Increased $CO_2$ concentration inside the flasks enhanced growth and/or rooting of several plants cultured in SSM or LM by increasing photosynthesis [7,29]. High $CO_2$

concentration promoted the growth of shoots of *Eucalyptus tereticornis* Sm. [30], *Celastrus paniculatus* Wall. [31] and *Vernonia condensata* Baker [32] supplemented with conventional sugar concentrations and allowed the growth of shoots cultured with low sucrose or even without any supplementary carbohydrate, as reported for *Paulownia fortunei* (Seem.) Hemsl. [33], *Samanea saman* (Jacq.) Merr. [34], several species of *Eucalyptus* [35–38] and macadamia [39]. In those studies, $CO_2$ concentration was increased by natural ventilation (by using plant containers with caps with permeable filters and placed in a $CO_2$-enriched chamber) or by forced ventilation (by the injection of sterile $CO_2$-enriched air into the culture vessels at fixed intervals). In the experiment described in Figure 3, $CO_2$-enriched air entered the bioreactors during immersion, while jars did not receive $CO_2$ supplementation. For willow, the use of $CO_2$-enriched air favoured photomixotrophic propagation and was decisive for photoautotrophic propagation, as no proliferation was observed in shoots cultured without sucrose in bioreactors which only received ambient $CO_2$ (Figure 1g).

In our experiments, we tried to minimise any possible carryover of sugars present in the tissues of the initial explants that had been conventionally cultured with 3% sucrose. First, we used apical sections because they are more uniform and are less likely to have stored carbohydrates or any other compounds in their tissues that could interfere with their response to sugar deprivation. Second, we performed two successive subcultures before recording the data, so that in the first subculture of each series the explants were already submitted to the same sucrose treatments as in the second, and thus, carryover of sugars could be considered negligible. After demonstrating that willow could be cultured without sugar by TIS, suggesting an increase in photosynthesis with this system, we included basal sections as initial explants, as they had produced more yield in our previous report [6]. Not surprisingly, basal sections showed significantly more proliferation capacity than apical sections, especially for the shoots cultured without sucrose. Basal segments of shoots from other trees such as chestnut were successfully used for culture in SSM and LM, both under photomixotrophic [40] and photoautotrophic conditions [41]. However, despite their larger size and putatively higher capacity for nutrient storage, basal sections did not grow enough to survive when cultured in jars without sugar (Figure 1f), reinforcing the importance of a proper gaseous exchange to attain autotrophy in micropropagation [7,29,42].

In the first experiments of this study, we observed that most of the leaves of shoots cultured in jars with low sugar were chlorotic (Figure 1e,f). In the following experiments we collected leaves to quantify the photosynthetic pigment content. As expected, shoots cultured in jars with less than 3% sucrose showed a considerable decrease in chlorophylls and carotenoids and less growth than the rest of the treatments (Figure 3). However, it was only a weak connection between the concentration of photosynthetic pigments and the growth parameters of willow shoots, as similar pigment concentrations were found in all the shoots cultured by TIS and the explants cultured in jars with 3% sucrose, whereas their growth response was significantly different. Among the shoots cultured in bioreactors, the pigment concentrations also showed no correlation with the sucrose supplementation.

Lack of correlation between photosynthetic pigments and growth has been reported for other plants such as myrtle [43], chestnut [44], apple [28], tobacco, potato, strawberry, and rapeseed [45] and *V. condensata* [32]. Even if chlorophylls and carotenoids drive photosynthesis by absorbing energy from light, photosynthesis is dependent on other major physiological and biochemical processes including $CO_2$ stomatal and mesophyll conductance and the biochemistry driven by the Rubisco enzyme system [46,47]. Net $CO_2$ assimilation depends, in turn, on the balance between photosynthesis, photorespiration and mitochondrial respiration [48], and finally, growth is an even more complex trait influenced by photosynthesis and other factors like nutrient availability, plant growth regulators, water relations and environmental factors. Regarding micropropagated plants, some of the key factors influencing photosynthesis have been studied in shoots cultured with or without exogenous sugar supplementation [49–51]. It has been reported that Rubisco decreased in avocado micropropagated with high sucrose content even when grown with $CO_2$-enriched air [52], but it seems that the concentration of this enzyme in micropropagated plants is

species-dependent [53], as reported for other biochemical parameters [45]. In our case, all the treatments with high proliferation exhibited high pigment content, but we cannot attribute the improvement in the yield of willow shoots cultured in TIS without sugar specifically to a superior content in photosynthetic pigments.

In our previous report on willow micropropagation with 3% sucrose [6], willow shoots formed adventitious roots spontaneously during the multiplication step. In the present study, root formation was dependent on the sucrose supplementation for both jars and bioreactors. Roots developed in SSM with 3% sucrose were abundant and well developed, whereas those formed with low sucrose treatments were scarce, small and frail. Interestingly, willow shoots cultured photoautotrophically in LM devoid of sucrose exhibited the highest amount of aerial biomass but the lowest root biomass, which increased when sucrose was added to the medium. Similarly, a correlation between sucrose supplementation, root system development and survival has been reported for other plants cultured by TIS such as Chinese ash [25], poplar [54] and bamboo [19]. In papaya, using a stationary liquid culture with zeolite supported more rooting and the survival of shoots cultured without sugar and exposed to high ventilation was reported [55]. Within plants cultured in SSM results seem to be more variable: grapevine and kiwi fruit shoots cultured in SSM rooted and acclimatized better with sucrose [56,57], in rapeseed no correlation could be established [45], whereas with thyme, paulownia, potato, tobacco and wasabi more roots were formed when less sugar was added to the culture medium [29,33,45,58]. In strawberry, contradictory reports have been reported, as Ševčíková et al. [45] did not observe an increase in the root mass when sugar was not added to the medium, whereas Nguyen et al. [29] reported more and longer roots in photoautotrophic conditions.

In the present study, the shoots cultured in SSM with low concentration of sucrose hardly produced any roots, and the shoots were too weak to undergo acclimatization, whereas in bioreactors, the extent of the root system did not limit the acclimatization ability of shoots cultured with low or no sucrose, as all plantlets acclimatized successfully, irrespective of the sucrose supplementation and the aerial and root biomass shown during the multiplication stage.

## 5. Conclusions

The way sucrose influenced willow propagation was dependent on the culture system. Explants cultured in closed jars produced less growth as sugar concentration decreased and were markedly inferior to shoots cultured in bioreactors. Bioreactor-grown shoots grew better with no added sucrose in the medium, suggesting an increase in photosynthesis in TIS. This is the first study demonstrating the feasibility of culturing willow in a sucrose-free medium and the use of TIS with $CO_2$ enrichment to achieve photoautotrophy. This investigation highlights procedures that may have wider applications in the vegetative propagation of willows and other plants where suitable material may be scarce or difficult to propagate.

**Author Contributions:** Conceptualization, N.V., M.Á.B. and C.S.; formal analysis, D.G., S.V. and N.V.; investigation, D.G., S.V., N.V. and A.A.; writing—review and editing, D.G. and N.V.; funding acquisition, N.V., M.Á.B. and C.S. All authors have read and agreed to the published version of the manuscript.

**Funding:** This research was funded by Xunta de Galicia (Spain) (projects IN607A and Contrato Programa 2019-2020), by CYTED (P117RT0522) and by CSIC (COOPB20584).

**Institutional Review Board Statement:** Not applicable.

**Informed Consent Statement:** Not applicable.

**Acknowledgments:** We thank Beatriz Cuenca (TRAGSA) for her valuable comments on photoautotrophic propagation, Mª José Cernadas for technical assistance and Bruce Christie for critical reading of the manuscript.

**Conflicts of Interest:** The authors declare no conflict of interest. The funders had no role in the design of the study; in the collection, analyses, or interpretation of data; in the writing of the manuscript, or in the decision to publish the results.

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
