# Peer review of "The Effect of Sucrose Supplementation on the Micropropagation of Salix viminalis L. Shoots in Semisolid Medium and Temporary Immersion Bioreactors"

_forests, doi:10.3390/f12101408_

Round 1
Reviewer 1 Report
Dear Authors,
Congratulations!
The manuscript is well-structured, has a great flow of ideas, and brings new information concerning the issues of the micropropagation of willow.
I suggest you try in a future experiment, apical explant with excised top
Author Response
We acknowledge the comments of reviewer 1 and appreciate the suggestions for next experiments.
Reviewer 1
Dear Authors,
Congratulations!
The manuscript is well-structured, has a great flow of ideas, and brings new information concerning the issues of the micropropagation of willow.
I suggest you try in a future experiment, apical explant with excised top
Reviewer 2 Report
Review
Photoautotrophic micropropagation of Salix viminalis shoots in temporary immersion bioreactors.
Photoautotrophic micropropagation is a technique in which micropropagation is carried out without sugars and organic nutrients in the medium. No amount of organic nutrients is added to the medium from outside. Vitamins, growth regulators and even gelling agents are also not added to the medium. The major dependency of this technique is on photosynthesis. The growth rate of the plants can be increased by increasing the rate of photosynthesis of plants. The authors of this paper try to find the best conditions for micropropagation of willows in temporary immersion bioreactors with limited availability of sucrose 30, 10, 5, 0 g/l under standard conditions (ambient CO2 (~ 400 ppm) and conventional PPF (~ 50 μmol m-2 s-1) and photoautotrophic conditions as elevated CO2 (~ 2000 ppm) and high PPF (~ 150 μmol m-2 s-1).
The first thing that attracts attention is the title, which narrows down the topics discussed. In addition to photoautotrophic micropropagation of Salix viminalis shoots in temporary immersion bioreactors, the authors also describe and analyze other variants of willow cultivation, such as the addition of sucrose, the type of medium (semi-solid and liquid) and the growth conditions (CO2 and PPF). So, I suggest changing the title. This is even more so since the authors themselves write: Line 36 In this study, we investigated the effect of sucrose supplementation during micropropagation of a willow species (Salix viminalis L.), with shoots cultured in semi-solid medium and temporary immersion bioreactors.
Detailed comments are below.
Introduction
Cultivation with temporary immersion bioreactors works especially well for recalcitrant. Willow is an easy plant to propagate, why did the authors choose this tree species?
Line29 “The cost of gelling agents is eliminated”. Yes, this is true, but the initial investment is high, commercial bioreactors are expensive. More to the point are the advantages: preventing morphological and physiological disorders, shortening the cycle of in vitro plant propagation, increasing the survival rate of in vitro plants when transferred to an ex vitro environment. Let us not forget the disadvantages of photoautotrophic micropropagation. The application of this technique requires a good knowledge of the physical in vitro environment of the systems. The cost of CO2 enrichment or using gas permeable filter discs to increase the CO2 concentration in the vessels, lighting and cooling is higher. This makes this technique somewhat costly. The technique has limited application to the propagation of plants with C4 or CAM photosynthetic pathways.
Materials and methods
The description of the method is vague; we do not know how many plants were tested in each variant.
Line 59: "6 explants per jar", but how many jars?
Line 68: "each RITA® contained 12 explants" - how many bioreactors did the authors use?
I found this information in Fig. 3. "two replicate experiments, each with 4 jars and 2 RITA® vessels".
Were the photoautotrophic conditions - CO2 levels (~ 2000 ppm) and high PPF (~ 150 μmol m-2 s-1) - also applied to the variant in the jars? If so, how were the vessels prepared for the treatment? Were the vessels placed in a CO2-enriched growth chamber? How was the ventilation performed?
Results
Line 197 : Fig. 1." Apical sections cultured in jars under high light intensity without sucrose (i, j) or with 3 g/L (?) sucrose (k)." - Or perhaps 30g/L. Was only high intensity light used for cultivation in jars or was CO2 enriched air used?
Line 201 : “The results of this experiment can be observed in Figure 2. Data were analyzed separately for each sucrose supplementation. The number of shoots was significantly affected by the growth conditions (p < 0.001 for both sucrose levels), with the highest proliferation obtained in PAM in both cases (Figure 2a)”. Photoautotrophic conditions (PAM) were defined as elevated CO2 (~ 2000 ppm) and high PPF (~ 150 μmol m-2 s-1). Photoautotrophy is a natural mode of plant nutrition. However, this mode of nutrition is suppressed in plants in which photomixotrophy is induced under tissue culture laboratory conditions and the provision of finished carbon in the form of sugars in the nutrient medium. Meanwhile, the authors used sucrose supplementation at a concentration of 10 and 30 g / l simultaneously. Are we dealing with photoautotrophy or photomixotrophy.
Line 273 : “Fig. 3 Effect of culture system and sucrose supplementation on proliferation rates (a-d) and pigment content (e, f) of apical sections of willow shoots grown under photoautotrophic conditions (150 μmol m-2s-1 PPF and 2000 ppm CO2 supplementation and 1 min immersion every 4 h in the case of bioreactors. “ Rather, we are dealing with photomixotrophy conditions, since sucrose was used.
Line 477:” shoots cultured in jars with less than 30 g/L sucrose showed a considerable decrease in chlorophylls and carotenoids and less growth than the rest of treatments (Figure 3)”.
Chlorophyll content indicates that plants are capable of photosynthesis. They could not do it without a carbon source. It is obvious
Concusion
Line 31: “The use of TIS with CO2 supplementation was crucial for the photoatotrophic micropropagation of willow shoots. Explants cultured in closed jars couldn’t resist sucrose deprivation, whereas those cultured in bioreactors without sucrose grew better than when this carbohydrate was added to the medium”.
I am not convinced that the methods used allow conclusions to be drawn about the study.
When explants in closed jars receive an addition of CO2? It would be good to study the parameters related to the intensity of photosynthesis.
Author Response
We acknowledge the comments and suggestions of reviewer 2 to improve the manuscript. Our answers are in italics
Photoautotrophic micropropagation of Salix viminalis shoots in temporary immersion bioreactors.
Photoautotrophic micropropagation is a technique in which micropropagation is carried out without sugars and organic nutrients in the medium. No amount of organic nutrients is added to the medium from outside. Vitamins, growth regulators and even gelling agents are also not added to the medium. The major dependency of this technique is on photosynthesis. The growth rate of the plants can be increased by increasing the rate of photosynthesis of plants. The authors of this paper try to find the best conditions for micropropagation of willows in temporary immersion bioreactors with limited availability of sucrose 30, 10, 5, 0 g/l under standard conditions (ambient CO2 (~ 400 ppm) and conventional PPF (~ 50 μmol m-2 s-1) and photoautotrophic conditions as elevated CO2 (~ 2000 ppm) and high PPF (~ 150 μmol m-2 s-1).
The first thing that attracts attention is the title, which narrows down the topics discussed. In addition to photoautotrophic micropropagation of Salix viminalis shoots in temporary immersion bioreactors, the authors also describe and analyze other variants of willow cultivation, such as the addition of sucrose, the type of medium (semi-solid and liquid) and the growth conditions (CO2 and PPF). So, I suggest changing the title. This is even more so since the authors themselves write: Line 36 In this study, we investigated the effect of sucrose supplementation during micropropagation of a willow species (Salix viminalis L.), with shoots cultured in semi-solid medium and temporary immersion bioreactors.
We changed the title as suggested by reviewer 2 to “The effect of sucrose supplementation on the micropropagation of Salix viminalis shoots in semisolid medium and temporary immersion bioreactors”
Detailed comments are below.
Introduction
Cultivation with temporary immersion bioreactors works especially well for recalcitrant. Willow is an easy plant to propagate, why did the authors choose this tree species?
We chose this species for two main reasons: i) Even if willows are easy to propagate as they root easily, for some applications as breeding it is advantageous to increase proliferation, as we did in the present study, ii) As a model species for developing efficient protocols for photoautotrophic propagation of other woody plants.
Now we have included that in the introduction section (Lines 59-70).
Line29 “The cost of gelling agents is eliminated”. Yes, this is true, but the initial investment is high, commercial bioreactors are expensive. More to the point are the advantages: preventing morphological and physiological disorders, shortening the cycle of in vitro plant propagation, increasing the survival rate of in vitro plants when transferred to an ex vitro environment. Let us not forget the disadvantages of photoautotrophic micropropagation. The application of this technique requires a good knowledge of the physical in vitro environment of the systems. The cost of CO2 enrichment or using gas permeable filter discs to increase the CO2 concentration in the vessels, lighting and cooling is higher. This makes this technique somewhat costly. The technique has limited application to the propagation of plants with C4 or CAM photosynthetic pathways.
We appreciate the comments of reviewer 2 and have made some changes in Lines 29-35 to include these ideas.
Materials and methods
The description of the method is vague; we do not know how many plants were tested in each variant.
Line 59: "6 explants per jar", but how many jars?
Line 68: "each RITA® contained 12 explants" - how many bioreactors did the authors use?
I found this information in Fig. 3. "two replicate experiments, each with 4 jars and 2 RITA® vessels".
We included the information about the number of jars and RITA® per experiment in the first description of the experiments in Materials and Methods section.
Were the photoautotrophic conditions - CO2 levels (~ 2000 ppm) and high PPF (~ 150 μmol m-2 s-1) - also applied to the variant in the jars? If so, how were the vessels prepared for the treatment? Were the vessels placed in a CO2-enriched growth chamber? How was the ventilation performed?
In the experiments described in Figure 3, jars were placed under high PPF but did not receive forced ventilation. We modified the description of the jars (line 80) to indicate that, and also the description of PAM conditions to clarify how this experimental unit works (lines 114-118), and added two pictures to Figure 1 (1a, b) to show the way jars were closed and how the CO2 enriched air enter in the bioreactors. In lines 124-126 we explain now that jars didn’t receive forced aeration.
Results
Line 197 : Fig. 1." Apical sections cultured in jars under high light intensity without sucrose (i, j) or with 3 g/L (?) sucrose (k)." - Or perhaps 30g/L. Was only high intensity light used for cultivation in jars or was CO2 enriched air used?
We are grateful to reviewer 2 for noticing this mistake (it was 30 g/L), now corrected. As said before, jars received high light but not forced ventilation.
Line 201 : “The results of this experiment can be observed in Figure 2. Data were analyzed separately for each sucrose supplementation. The number of shoots was significantly affected by the growth conditions (p < 0.001 for both sucrose levels), with the highest proliferation obtained in PAM in both cases (Figure 2a)”. Photoautotrophic conditions (PAM) were defined as elevated CO2 (~ 2000 ppm) and high PPF (~ 150 μmol m-2 s-1). Photoautotrophy is a natural mode of plant nutrition. However, this mode of nutrition is suppressed in plants in which photomixotrophy is induced under tissue culture laboratory conditions and the provision of finished carbon in the form of sugars in the nutrient medium. Meanwhile, the authors used sucrose supplementation at a concentration of 10 and 30 g / l simultaneously. Are we dealing with photoautotrophy or photomixotrophy.
We have modified the redaction of this point, using the word “photoautotrophy” later in the document and only for the treatments without sucrose.
Line 273 : “Fig. 3 Effect of culture system and sucrose supplementation on proliferation rates (a-d) and pigment content (e, f) of apical sections of willow shoots grown under photoautotrophic conditions (150 μmol m-2s-1 PPF and 2000 ppm CO2 supplementation and 1 min immersion every 4 h in the case of bioreactors. “ Rather, we are dealing with photomixotrophy conditions, since sucrose was used.
We have modified the redaction of this section, without using “photoautotropic conditons”. Instead, we say now that the explants were grown under high PPF and received CO2-enriched air.
Line 477:” shoots cultured in jars with less than 30 g/L sucrose showed a considerable decrease in chlorophylls and carotenoids and less growth than the rest of treatments (Figure 3)”.
Chlorophyll content indicates that plants are capable of photosynthesis. They could not do it without a carbon source. It is obvious
We also expected low concentrations of pigments in these treatments. We were interested in study a possible correlation between pigments and growth, but as we discuss later (now lines 539 and below) we couldn’t find it.
Concusion
Line 31: “The use of TIS with CO2 supplementation was crucial for the photoatotrophic micropropagation of willow shoots. Explants cultured in closed jars couldn’t resist sucrose deprivation, whereas those cultured in bioreactors without sucrose grew better than when this carbohydrate was added to the medium”.
I am not convinced that the methods used allow conclusions to be drawn about the study.
When explants in closed jars receive an addition of CO2? It would be good to study the parameters related to the intensity of photosynthesis.
We appreciate the suggestions of reviewer and will try to measure the actual photosynthetic ability of plants in next studies.
Reviewer 3 Report
I have revised the manuscript “Photoautotrophic micropropagation of Salix viminalis shoots in temporary immersion bioreactors" carefully. The topic of micropropagation of alternative plants has been very popular lately. The interest in in vitro techniques, especially with the use of alternative plants, has increased in recent years due to the wide possibilities of their use. In conventional cultivation conditions, obtaining a large number of plants is often difficult due to the low germination of seeds and their low vigor. The optimal solution in this situation is the use of a properly selected culturing in vitro.
Below, I am presenting my comments of the manuscript:
General comments:
Abstract:
line 20: line 20 and in the all place the article, please indicate the sucrose content in the medium in % or change g/L on g·dm-3
Introduction:
I think it is worth expanding the introduction a bit (about half a page more). It would be good for the authors to provide a brief description of the species and write to which botanical family Salix viminalis belongs. I also suggest that you include information on the reproduction of S. viminalis in natural conditions, and justify why research is being conducted on the micropropagation in vitro of this species?
Materials and Methods
Authors should include information on what statistical program in the reference list.
Results
line 191. Figure 1. lack of dot
In Figure 2, Figure 3… (lines 227, 267….): and in the all places this article Autors should change the unit µg/g on µg·g-1
Author Response
We acknowledge the comments and suggestions of reviewer 3 to improve the manuscript. Please find our responses in italics
I have revised the manuscript “Photoautotrophic micropropagation of Salix viminalis shoots in temporary immersion bioreactors" carefully. The topic of micropropagation of alternative plants has been very popular lately. The interest in in vitro techniques, especially with the use of alternative plants, has increased in recent years due to the wide possibilities of their use. In conventional cultivation conditions, obtaining a large number of plants is often difficult due to the low germination of seeds and their low vigor. The optimal solution in this situation is the use of a properly selected culturing in vitro.
Below, I am presenting my comments of the manuscript:
General comments:
Abstract:
line 20: line 20 and in the all place the article, please indicate the sucrose content in the medium in % or change g/L on g·dm-3
We have changed g/L to % throughout the manuscript
Introduction:
I think it is worth expanding the introduction a bit (about half a page more). It would be good for the authors to provide a brief description of the species and write to which botanical family Salix viminalis belongs. I also suggest that you include information on the reproduction of S. viminalis in natural conditions, and justify why research is being conducted on the micropropagation in vitro of this species?
We have expanded the introduction as suggested by Reviewer 3, including the main reasons for choosing this species: i) Even if willows are easy to propagate as they root easily, for some applications as breeding it is advantageous to increase proliferation, as we did in the present study, ii) As a model species for developing efficient protocols for photoautotrophic propagation of other woody plants.
Materials and Methods
Authors should include information on what statistical program in the reference list.
Done
Results
line 191. Figure 1. lack of dot
We are grateful to Reviewer 3 for noticing this mistake, now corrected.
In Figure 2, Figure 3… (lines 227, 267….): and in the all places this article Autors should change the unit µg/g on µg·g-1
Done
Round 2
Reviewer 2 Report
The article does not address the effect of sucrose in either the introduction or the discussion. So please reconsider the topic and the title.
I propose to consider the issues the authors wish to discuss in this article entitled: The effect of sucrose supplementation on the micropropagation of Salix viminalis shoots in semisolid medium and in temporary immersion bioreactors.
The attempts to introduce photoautotrophic cultures on both solid media and aggregates are interesting. The authors aimed to use S. viminalis as a model to develop an efficient protocol for the cultivation of willow axillary shoots without sucrose addition.
Growth conditions (light intensity and CO2 addition), frequency of immersion, type of explant, and sucrose addition were evaluated in semisolid and liquid medium using RITA® bioreactors. In their study, the authors describe the effects of culture system and sucrose addition, type of explant and frequency of immersion on shoot quality, photosynthetic pigment content, in vitro proliferation rate and ex vitro performance.
Author Response
Reviewer 2
We acknowledge the comments and suggestions of reviewer 2 to improve the manuscript. Our answers are in italics
Comments and Suggestions for Authors
The article does not address the effect of sucrose in either the introduction or the discussion. So please reconsider the topic and the title.
In the first round of revision we modified the tittle following the suggestion of Reviewer2. For this reason, we infer now that in this second round the Reviewer2’s comments are aimed to better focus the writing of the article on the sucrose effect. We added some content in the introduction, discussion and conclusion sections to make more visible the ideas we had been discussing.
I propose to consider the issues the authors wish to discuss in this article entitled: The effect of sucrose supplementation on the micropropagation of Salix viminalis shoots in semisolid medium and in temporary immersion bioreactors.
We modified the discussion section to make clearer now that we focus on the effect of sucrose on the proliferation and rooting of willow shoots in jars and bioreactors (effect of the culture system), the influence of the growth conditions on the capacity of willow shoots for growing with low or no sucrose to achieve photoautotrophy (the interrelation between sucrose supplementation, CO2 availability and growth), the relation between growth, sucrose and photosynthetic pigments, and the acclimation capacity of willow shoots cultured without sucrose, which in the case of willow seems not to depend on the size of the root system before ex vitro transference.
The attempts to introduce photoautotrophic cultures on both solid media and aggregates are interesting. The authors aimed to use S. viminalis as a model to develop an efficient protocol for the cultivation of willow axillary shoots without sucrose addition.
We appreciate the Reviewer comment about the interest of our study.
Growth conditions (light intensity and CO2 addition), frequency of immersion, type of explant, and sucrose addition were evaluated in semisolid and liquid medium using RITA® bioreactors. In their study, the authors describe the effects of culture system and sucrose addition, type of explant and frequency of immersion on shoot quality, photosynthetic pigment content, in vitro proliferation rate and ex vitro performance.
We agree with the Reviewer comment about the contents of our study.